

# moSCminer: a cell subtype classification framework based on the attention neural network integrating the single-cell multi-omics dataset on the cloud

Joung Min Choi[1,*], Chaelin Park[2,*] and Heejoon Chae[2]

[1] Department of Computer Science, Virginia Polytechnic Institute and State University (Virginia Tech), Blacksburg, Virginia, United States
[2] Division of Computer Science, Sookmyung Women's University, Seoul, South Korea
* These authors contributed equally to this work.

Corresponding author
Heejoon Chae,
heechae@sookmyung.ac.kr

## ABSTRACT

Single-cell omics sequencing has rapidly advanced, enabling the quantification of diverse omics profiles at a single-cell resolution. To facilitate comprehensive biological insights, such as cellular differentiation trajectories, precise annotation of cell subtypes is essential. Conventional methods involve clustering cells and manually assigning subtypes based on canonical markers, a labor-intensive and expert-dependent process. Hence, an automated computational prediction framework is crucial. While several classification frameworks for predicting cell subtypes from single-cell RNA sequencing datasets exist, these methods solely rely on single-omics data, offering insights at a single molecular level. They often miss inter-omic correlations and a holistic understanding of cellular processes. To address this, the integration of multi-omics datasets from individual cells is essential for accurate subtype annotation. This article introduces moSCminer, a novel framework for classifying cell subtypes that harnesses the power of single-cell multi-omics sequencing datasets through an attention-based neural network operating at the omics level. By integrating three distinct omics datasets—gene expression, DNA methylation, and DNA accessibility—while accounting for their biological relationships, moSCminer excels at learning the relative significance of each omics feature. It then transforms this knowledge into a novel representation for cell subtype classification. Comparative evaluations against standard machine learning-based classifiers demonstrate moSCminer's superior performance, consistently achieving the highest average performance on real datasets. The efficacy of multi-omics integration is further corroborated through an in-depth analysis of the omics-level attention module, which identifies potential markers for cell subtype annotation. To enhance accessibility and scalability, moSCminer is accessible as a user-friendly web-based platform seamlessly connected to a cloud system, publicly accessible at http://203.252.206.118:5568. Notably, this study marks the pioneering integration of three single-cell multi-omics datasets for cell subtype identification.

## INTRODUCTION

Recent strides in single-cell omics sequencing technologies have ushered in a new era of cellular exploration, offering insights into developmental stages, cellular phenotypes, and pathogenesis (*Haghverdi & Ludwig, 2023*; *Li & Wang, 2021*; *Nomura, 2021*). By delving into the profiles of various modalities such as transcriptome and epigenome, single-cell studies empower precise profiling at the individual cell level. This sharpens our understanding compared to bulk sequencing methods that blend data from millions of cells, masking the nuances between cell subtypes (*Adossa et al., 2021*). In this context, accurate cell subtype identification has emerged as a pivotal requirement for in-depth research into tissue heterogeneity, complex differentiation, and disease-related developmental strategies at the cellular level (*Shalek et al., 2014*). Traditionally, cell subtype prediction has relied heavily on single-cell RNA sequencing datasets, often employing unsupervised learning-based approaches (*Zhang et al., 2023*). These methods typically embark on clustering-based pipelines, reducing dataset dimensionality to distill low-dimensional representations, conducting clustering to identify distinct cell groups, and assigning cell subtypes through manual examination involving canonical cell subtype-specific marker genes (*Luecken & Theis, 2019*). Yet, these approaches grapple with a significant drawback: their reliance on extensive knowledge of various cell populations and marker genes, entailing a labor-intensive, less reproducible process (*Nguyen & Griss, 2022*).

Recent years have witnessed the emergence of supervised-learning-based methods for automating cell subtype prediction (*De Kanter et al., 2019*; *Lin et al., 2020*; *Nguyen & Griss, 2022*). These methods leverage machine learning algorithms or neural networks, training models to learn cell subtype classification weights or parameters independently of marker genes or manual inspection. While these methods exhibit promising performance, those rooted solely in single-omics datasets harness information from a solitary molecular level. Accumulating evidence suggests that multi-omics profiling offers more robust cell subtype classification, as parallel profiles from multiple layers unveil cell subtype-specific networks spanning various biological processes, such as epigenetic regulation and gene expression (*Bai, Peng & Yi, 2021*). The integration of single-cell RNA sequencing (scRNA-seq) and single-cell assay for transposase-accessible chromatin sequencing (scATAC-seq) has facilitated the comprehensive characterization of critical transcription factors and regulatory elements underlying various human cancers. Notably, this approach has been applied to investigate clear cell renal cell carcinoma (*Long et al., 2022*), colorectal cancers (*Zhu et al., 2023*), and breast cancers (*Zhu et al., 2023*), shedding light on the intricate molecular landscape of these malignancies. The simultaneous profiling of multiple single-cell omics data types has revealed distinct differentiation states within gastric cancer. By elucidating the relationships between genetic lineages, DNA methylation

patterns, and transcriptomic clusters at the single-cell level, researchers have gained valuable insights into the heterogeneity of gastric cancer (*Bian et al., 2023*). Additionally, the scWGS-RNA-seq method, designed to amplify single-cell DNA and RNA without separating them, has proven instrumental in detecting unique cell subpopulations, particularly in the context of true normal cells. By harnessing information from both the genome and transcriptome, this approach, exemplified by the study conducted by *Yu et al. (2023)*, showcases the potential to unravel previously unseen cellular diversity and heterogeneity. Nonetheless, integrating single-cell multi-omics datasets remains a challenge due to the high dimensionality inherent to each dataset. Approaches to address this challenge, including clustering methods and non-negative matrix factorization, have been widely explored (*Eltager et al., 2022*; *Taguchi & Turki, 2021*). Recent studies have introduced neural network-based strategies for single-cell multi-omics data integration, capturing informative nonlinear features within the latent space (*Leng et al., 2022*; *Lin et al., 2022*). Attention neural network has been adopted for bulk sequencing-based multi-omics profiles to learn the feature's relationship and the integrated representations. MOMA has been introduced as disease classification method based on a multi-task attention learning algorithm for two omics data integration and verified its utility for biological analysis (*Moon & Lee, 2022*). SADLN, on the other hand, utilized a self-attention mechanism to train and learn integrated latent features from multi-omics datasets. These features were subsequently employed as input for a Gaussian Mixture model to discern cancer subtypes effectively (*Sun et al., 2023*). MOCDN presented self-attention-based neural network model to integrate three different omics profiles and identified biomarkers of kidney renal cell carcinoma (*Gong et al., 2023*). These strategies have delved into the causal factors governing cellular states, delivering promising results by unveiling potential regulatory influences. Yet, the widespread adoption of multi-omics integration for cell subtype prediction remains limited.

In this article, we propose moSCminer, a web-based cloud platform for cell subtype classification framework integrating the single-cell multi-omics dataset based on the omics-level attention neural network. Preprocessing and feature selection were performed based on the transformation of each omics dataset to a gene-based matrix, considering the biological interplay across gene expression, DNA methylation, and DNA accessibility. To integrate multi-omics more efficiently by reducing the dimensionality of each omics, not ignoring the distribution difference of each omics dataset, self-attention mechanism was employed to each preprocessed omics dataset. Each feature was transformed to new representations, factoring in their relative importance. Features from each omics were then concatenated and delivered to the fully connected layers to predict the subtype of each cell. Benchmarking moSCminer against various machine learning-based classifiers, our model consistently outperformed the rest, boasting the highest average accuracy and weighted F1-score across real datasets. In addition, our experiments show that omics-level attention improves the prediction performance with the identification of the marker candidates to distinguish the cell subtypes. To the best of our knowledge, this is the first study integrating three single-cell multi-omics datasets for the cell subtype classification, showing the improvement of prediction compared to the usage of single-omics.
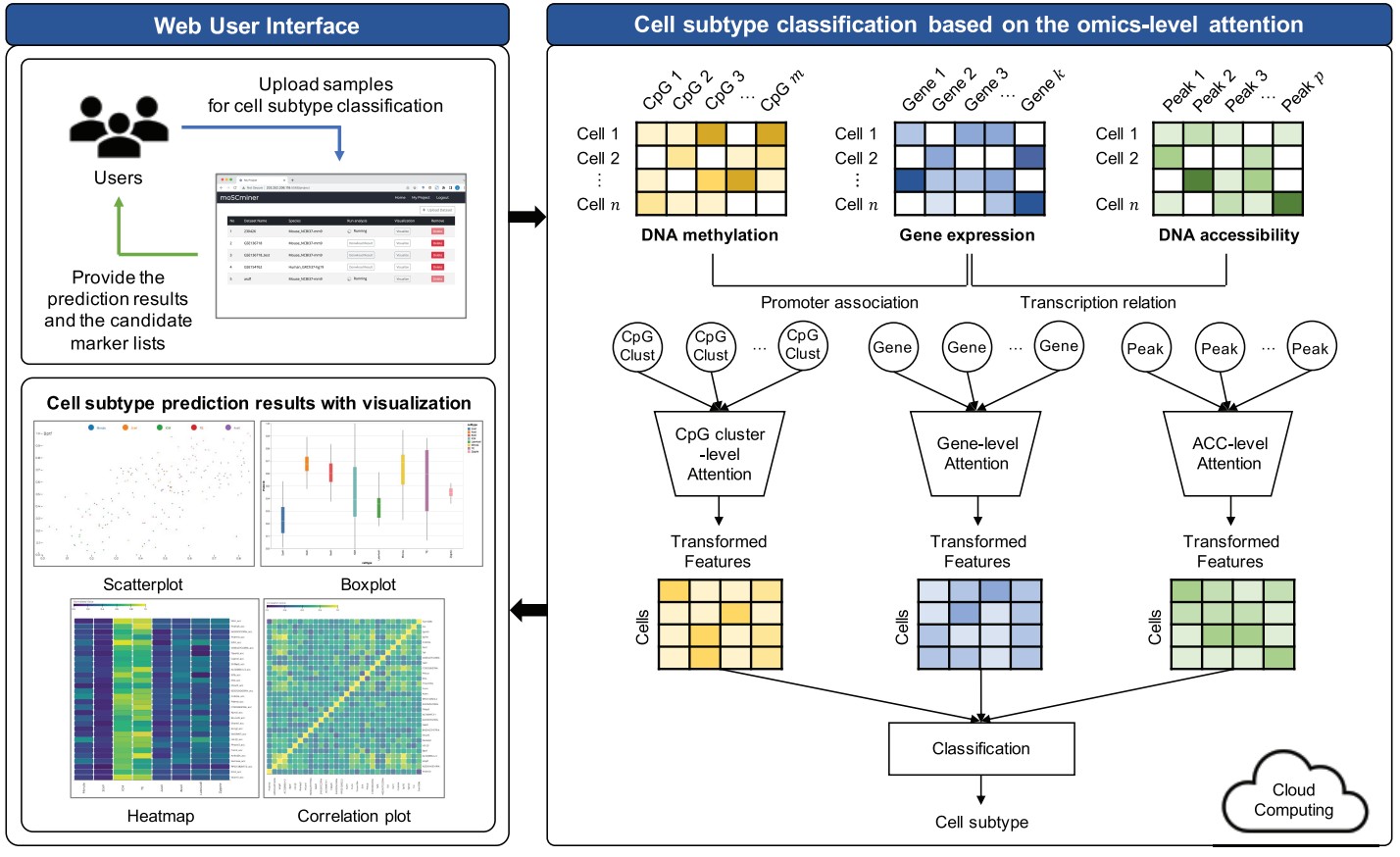

**Figure 1 Workflow illustrating the proposed cell subtype prediction model based on the integration of single-cell multi-omics data.**

To enhance user efficiency in the experimental setting, we developed moSCminer as an interactive web-based platform featuring intuitive interfaces that eliminate the need for software installation. Users can upload their single-cell multi-omics datasets, and our platform will automatically run our modules and provide the cell subtype annotation results. For a comprehensive analysis of their single-cell multi-omics studies, we offer visualizations of cell subtype classification results, identified marker candidate lists, and relevant biological information—all easily accessible. To address computational cost limitations, our platform is integrated with a cloud system, ensuring scalability for analytical processes.

## METHODS

The proposed model comprises three key steps: (1) Preprocessing, (2) feature selection, and (3) cell subtype classification utilizing omics-level attention. The workflow of this model is visually represented in Fig. 1.

### Preprocessing and feature selection

To effectively integrate a multi-omics dataset, the conventional approach involves converting the sparse DNA methylation and DNA accessibility dataset matrices into

gene-based matrices (*Taguchi & Turki, 2021*; *Xu, Begoli & McCord, 2022*). Here's a breakdown of our preprocessing steps:

For gene expression data, we initially removed genes that lacked read counts for all samples. Subsequently, read counts were normalized according to library size and log-transformed using the 'Scanpy' Python package (*Wolf, Angerer & Theis, 2018*).

Concerning the DNA methylation dataset, we adopted a strategy where CpGs within 2 kb of the promoter regions of each gene were grouped to form a cluster (referred to as CpG cluster). We calculated the cluster's average methylation values. This approach aligns with existing research demonstrating that the distribution of CpG density around promoter regions is closely linked to gene transcription and expression regulation (*Deaton & Bird, 2011*; *Tian et al., 2022*). To mitigate bias stemming from frequent missing values during model training, we performed mean imputation, eliminating CpG clusters with missing values for all samples.

In the case of DNA accessibility, we considered the summation of accessibility values for each gene body, which may be associated with transcription (*Xu, Begoli & McCord, 2022*). To achieve this, we summed all accessibility peaks within each transcription region to represent gene activity.

Subsequent to preprocessing, features lacking a common gene-based relationship across any omics were filtered out. This was essential to prevent overfitting resulting from high dimensionality, while simultaneously facilitating multi-omics integration grounded in biological connections. Min-max normalization was conducted on the gene expression and DNA accessibility datasets to align their value ranges with that of the DNA methylation dataset.

## Cell subtype classification based on the omics-level attention

To empower neural networks for more effective cell subtype classification, we harnessed a self-attention mechanism. This mechanism trains the model to discern the relative importance of each feature (*Lin et al., 2017*). The self-attention approach was applied individually to each omics dataset. The model subsequently reconstructed features based on their learned importance weight for each omics. These features from all omics datasets were concatenated, creating a new feature representation for cell subtype classification.

Let $k$ represent the number of features, $x_i \in \mathbf{R}^k$ denote the $i$th sample, and $x = (x_1, \ldots, x_n) \in \mathbf{R}^{n \times k}$ represent a matrix containing all $x_i$. For each feature $j \in \{1:k\}$, we generated a $m$-dimensional embedding vector $e_j$ using random vectors to represent $x_i$ to $\widehat{x}_i$ through multiplication (*Beykikhoshk et al., 2020*).

$$\hat{x}_i^{(j)} = f_e\left(e_j, x_i^{(j)}\right) = e_j x_i^{(j)}, \tag{1}$$

The model assigns an attention score, $\alpha_j$, to each feature $j$ as follows:

$$\bar{x}_i^{(j)} = tanh\left(W_1 \hat{x}_i^{(j)} + b_1\right) \tag{2}$$

$$s_i^{(j)} = W_3 tanh\left(W_2 \hat{x}_i^{(j)} + b_2\right) \tag{3}$$

$$\alpha_i^{(j)} = \frac{\exp\left(s_i^{(j)}\right)}{\sum_{l=1}^{k} \exp\left(s_i^{(l)}\right)} \tag{4}$$

$$c_i^{(j)} = \sum_{j=1}^{k} \alpha_i^{(j)} \bar{x}_i^{(j)}, \tag{5}$$

where $W_1$, $W_2$, $W_3$ are the weights and $b_1$ and $b_2$ are the bias terms for each layer. $s_i^{(j)}$ is the attention score representing the importance of each feature $\hat{x}_i^{(j)}$ for the $i$th sample, which is converted to $\alpha_i^{(j)}$ *via* normalization using the softmax function. Based on this, $\hat{x}_i^{(j)}$ is transformed into a new feature representation of $c_i^{(j)}$ by the weighted sum of the encoded feature vectors $\bar{x}_i^{(j)}$ and normalized attention scores $\alpha_i^{(j)}$. Each omics dataset underwent this self-attention mechanism. Transformed representations were then concatenated and forwarded to two fully connected layers, culminating in a softmax function for cell subtype classification.

To train the model, we utilized cross-entropy loss as the loss function:

$$\mathscr{L} = -\sum_{i=1}^{C} y_i \log(\hat{y}_i), \tag{6}$$

where $C$ denotes the number of cell subtypes, and $y$ and $\hat{y}$ represent the true and model-predicted subtype probability distributions, respectively. To prevent overfitting, we incorporated dropout in the fully connected layers.

### Implementation of web-based platform connected with the cloud system

We have developed moSCminer as a user-friendly web-based platform connected to a cloud system for enhanced user accessibility and convenience. The platform is designed to automate the analysis process and reduce user intervention. Here are the key features:

- User-friendly interface: The platform boasts a user-friendly interface accessible without requiring users to log in. Whether you have an account or prefer the guest mode, uploading your single-cell multi-omics datasets is a breeze.
- Automated cloud-based analysis: Our web-based platform handles the entire analysis pipeline, encompassing preprocessing, feature selection, and cell subtype classification. These computationally intensive tasks are executed on a cloud system, sparing users the need to manage complex computations.
- Email notifications: Users are promptly notified of their analysis results *via* email. These results encompass cell subtype annotations and lists of high-importance biomarker candidates identified during the prediction step.

- Interactive visualizations: We understand the significance of visualizing and interpreting results. Thus, our platform offers interactive visualizations of cell subtype classification results. Users can easily access and download various types of figures, including box plots, scatter plots, heatmaps, and correlation plots. These visual aids provide valuable insights into the data.
- Marker candidates: Our platform offers information on marker candidates, making it easier for users to distinguish cell subtypes effectively.
- Scalability: To address potential computational limitations, our platform is connected to a cloud system, ensuring scalability for analysis tasks of varying complexities.

The front-end of the website was meticulously crafted using HTML5, JavaScript, D3, JQuery, and CSS3. On the backend, a web server was developed using Node.js (*Tilkov & Vinoski, 2010*). The preprocessing, feature selection, and cell subtype classification modules were implemented using Python, utilizing libraries such as Tensorflow (*Abadi et al., 2015*) and Scikit-learn (*Pedregosa et al., 2011*). The Linux Bash Shell was also employed to facilitate these tasks.

In summary, moSCminer represents a powerful tool for single-cell multi-omics data analysis. Its user-friendly interface, automated cloud-based analysis, interactive visualizations, and scalability make it a valuable resource for researchers seeking to extract meaningful insights from complex datasets. Additionally, its ability to integrate multiple single-cell omics datasets offers improved cell subtype prediction, opening new avenues for understanding cellular heterogeneity and differentiation processes.

## RESULTS

### Experimental design

#### Dataset

To evaluate the performance of the proposed model, we acquired three publicly available single-cell multi-omics datasets from the Gene Expression Omnibus repository: GSE154762 (*Yan et al., 2021*), GSE136718 (*Wang et al., 2021*), and GSE140203 (*Ma et al., 2020*). The GSE154762 dataset comprised 899 single human oocytes and somatic cells having nine cell subtypes, obtained through scChaRM-seq. The GSE136718 dataset consisted of 210 cells with eight subtypes related to mouse embryo development, obtained through scNOMeRe-seq. Both datasets provided profiles for three omics types (gene expression, DNA methylation, and DNA accessibility), and information on cell subtypes for each sample. For the robust testing with the different scenarios, we also obtained adult mouse skin datasets composed of 32,321 cells with 22 cell subtypes from GSE140203, which provided gene expression and DNA accessibility data based on SHARE-seq. The number of samples and cell subtypes for each dataset are summarized in Table 1. For the number of features, the number of features varies greatly for each sample. GSE154762 dataset had 23,513 genes, while for DNA methylation, each sample had different number of features ranging from 4,233 to 12,500,793, and similar to DNA accessibility, having different number of features from 36,449 to 94,472,780. GSE136718 dataset consists of gene expression profiles with 24,963 genes, where for DNA methylation, samples showed

| Table 1 Number of samples for each cell subtype | | |
|---|---|---|
| **Dataset** | **Cell subtype** | **Number of samples** |
| GSE154762 | FGO | 81 |
| | GO1 | 40 |
| | GO2 | 46 |
| | Granulosa | 93 |
| | Immune | 20 |
| | MI | 155 |
| | MII | 90 |
| | StromaC1 | 189 |
| | StromaC2 | 185 |
| GSE136718 | 2 cell | 76 |
| | 4 cell | 67 |
| | 8 cell | 31 |
| | ICM | 36 |
| | Late4cell | 22 |
| | Morula | 24 |
| | TE | 21 |
| | Zygote | 12 |
| GSE140203 | ahighCD34+ bulge | 1,556 |
| | alowCD34+ bulge | 1,877 |
| | Basal | 7,787 |
| | Dermal Fibroblast | 1,121 |
| | Dermal Papilla | 766 |
| | Dermal Sheath | 398 |
| | Endothelial | 927 |
| | Granular | 291 |
| | Hair Shaft-cuticle-cortex | 1,166 |
| | Infundibulum | 4,139 |
| | IRS | 672 |
| | Isthmus | 689 |
| | K6+ Bulge companion layer | 514 |
| | Macrophage DC | 263 |
| | Medulla | 981 |
| | Melanocyte | 187 |
| | ORS | 1,029 |
| | Schwann cell | 163 |
| | Sebaceous gland | 181 |
| | Spinous | 3,146 |
| | TAC-1 | 3,370 |
| | TAC-2 | 1,008 |

the different number of CpGs from 659,456 5,729,653 and the number of sites from 5,807,264 to 51,503,457 for DNA accessibility. GSE140203 dataset had 23,297 genes and the number of sites from 1,000 to 101,593 for DNA accessibility. Through preprocessing and feature selection steps, GSE154762 dataset had total of 1,608 features (793 genes, 373 CpG clusters 442 accessibilities) and GSE136718 dataset consists of total of 303 features having 101 feature for each omics, respectively. Total of 762 features having 381 features for each omics were selected for GSE140203 dataset.

*Model optimization*

The hyperparameters of our proposed model were optimized for each dataset using a grid search approach. Each dataset was randomly split into training and testing sets at an 8:2 ratio. We selected the parameter combination with the best testing accuracy. We utilized the 'adam' optimizer (*Kingma & Ba, 2015*), with a learning rate of 1e−3, 300 training epochs, and a batch size of 128. From the optimization results presented in Table S1, we determined that the embedding vectors $e_j$ and $\bar{x}_i$ had sizes of 128 and 64, respectively. The number of hidden nodes in the fully connected layer was set at 100, and the dropout rate was 0.2.

## Performance evaluation with the baseline methods

To assess the effectiveness of our proposed method, we compared its performance to that of several machine learning-based classifiers, including Support Vector Machine (SVM), Random Forest (RF), Logistic Regression (LR), and Naive Bayes (NB), implemented using the 'Scikit-learn' package (*Pedregosa et al., 2011*). Similar to our model, we optimized each classifier for each dataset through grid search, selecting the hyperparameter combination yielding the highest average accuracy for the testing dataset. Those results are shown in Tables S2–S4. We performed five-fold cross-validation to evaluate accuracy, the weighted F1-score, the Matthews correlation coefficient (MCC), and the Area under the ROC Curve (AUC) as evaluation metrics. Additionally, we used the same multi-omics features selected during the feature selection step for the baseline methods. As illustrated in Fig. 2 and Table 2, moSCminer outperformed the baseline methods, achieving an average weighted F1-score of 0.954 and 0.986 for the GSE154762 and GSE136718 dataset, respectively, compared to the second-highest average F1-score of 0.916 and 0.970 achieved by RF. For the largest dataset of GSE140203, composed of 32,321 cells with 22 cell subtypes, our method achieved the highest classification performance with an average AUC of 0.983, where RF obtained 0.926. The cell subtype-wise performance results were reported in Table S5. For GSE136718 and GSE154762 datasets, relatively having small number of samples and subtypes, moSCminer generally demonstrated the best or similar prediction performance compared to other methods. But, when applied to more complex cell subtype prediction task using GSE140203 dataset, moSCminer outperformed all the baseline methods and its variant for all 22 cell subtypes, achieving the best performance.

## Effectiveness of omics-level attention

Our proposed method leverages omics-level attention to transform features into new representations, capturing the relative importance weights for cell subtype classification.

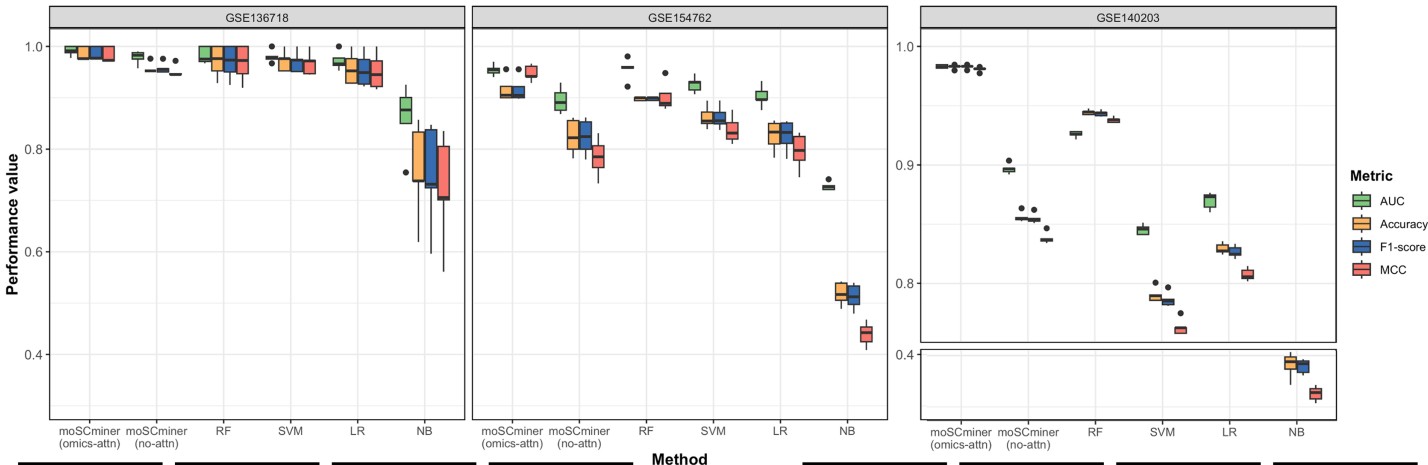

**Figure 2 Performance comparison of moSCminer with its variant and the baseline methods based on five-fold cross-validation.** A horizontal line within the box represents the median of performance values for each method.

**Table 2 Average performance results for cell subtype predictions of moSCminer with its variant and the baseline methods based on five-fold cross-validation.**

| Dataset | Metric | moSCminer (omics-attn) | moSCminer (no-attn) | RF | SVM | LR | NB |
|---|---|---|---|---|---|---|---|
| GSE136718 | Accuracy | **0.986** | 0.957 | 0.971 | 0.971 | 0.957 | 0.757 |
| | F1-score | **0.986** | 0.958 | 0.970 | 0.970 | 0.955 | 0.748 |
| | MCC | **0.983** | 0.951 | 0.968 | 0.967 | 0.951 | 0.722 |
| | AUC | **0.991** | 0.979 | 0.983 | 0.980 | 0.972 | 0.861 |
| GSE154762 | Accuracy | **0.956** | 0.816 | 0.917 | 0.862 | 0.827 | 0.518 |
| | F1-score | **0.954** | 0.817 | 0.916 | 0.862 | 0.826 | 0.512 |
| | MCC | **0.948** | 0.784 | 0.902 | 0.838 | 0.796 | 0.439 |
| | AUC | **0.977** | 0.895 | 0.933 | 0.926 | 0.903 | 0.728 |
| GSE140203 | Accuracy | **0.983** | 0.856 | 0.944 | 0.790 | 0.829 | 0.382 |
| | F1-score | **0.983** | 0.855 | 0.944 | 0.786 | 0.827 | 0.379 |
| | MCC | **0.981** | 0.838 | 0.938 | 0.763 | 0.808 | 0.326 |
| | AUC | **0.983** | 0.897 | 0.926 | 0.845 | 0.870 | 0.654 |

**Note:**
The bolded font was used to highlight the method that exhibited the highest performance among the various approaches.

To assess the impact of omics-level attention on prediction performance, we implemented a variant of our model by removing the attention components, referred to as "moSCminer (no-attn)," and conducted 5-fold cross-validation to compare performance. Notably, without the attention module, classification performance decreased (Fig. 2 and Table 2). For the GSE136718 dataset, the average accuracy dropped from 0.986 to 0.957, while for relatively having larger datasets with more samples and complex cell subtypes, a significant performance drop was observed for average accuracy, from 0.956 to 0.816 (GSE154762), and from 0.983 to 0.856 (GSE140203). Similar performance change was also shown in the cell subtype-wise performance results (Table S5). These results underscore the effectiveness of omics-level attention in our proposed method and the importance of new

**Table 3** Average cell subtype classification performance using different combinations of omics datasets.

| Number of omics | Dataset / Metric | GSE154762 | | GSE136718 | | GSE140203 | |
|---|---|---|---|---|---|---|---|
| | | Accuracy | F1-score | Accuracy | F1-score | Accuracy | F1-score |
| Multi-omics | Gene+Methyl+Acc | 0.917 | 0.916 | 0.986 | 0.986 | – | – |
| | Gene+Methyl | 0.871 | 0.871 | 0.891 | 0.878 | – | – |
| | Gene+Acc | 0.740 | 0.728 | 0.957 | 0.956 | 0.983 | 0.983 |
| | Methyl+Acc | 0.868 | 0.864 | 0.581 | 0.552 | – | – |
| Single omics | Gene | 0.697 | 0.690 | 0.962 | 0.962 | 0.768 | 0.760 |
| | Methyl | 0.841 | 0.837 | 0.452 | 0.357 | – | – |
| | Acc | 0.340 | 0.288 | 0.443 | 0.315 | 0.950 | 0.951 |

feature representations derived from feature transformation in cell subtype classification using single-cell multi-omics data.

## Cell subtype prediction improvement by multi-omics integration

To evaluate whether cell subtype classification based on multi-omics integration improved performance, we conducted five-fold cross-validation and compared our model's performance using either single omics or a combination of two omics datasets. We denoted gene expression as 'gene,' DNA methylation as 'methyl,' and DNA accessibility as 'acc.' As shown in Table 3, utilizing single-omics dataset could provide the performance higher than 0.9, for example, the average accuracy of 0.962 in GSE136718 using gene expression profiles, or 0.950 using DNA accessibility data in GSE140203. However, the proposed model, when trained using all three multi-omics datasets, consistently achieved the highest average prediction performance across all three datasets. This indicates that integrating omics datasets from different biological layers enhances the cell subtype classification model's accuracy compared to using single omics data.

## Identification of marker candidates for cell subtype prediction

During the training phase of moSCminer, the omics-level attention scores were learned, providing relative importance scores for cell subtype classification. Features with the highest attention scores hold the potential to be cell marker candidates for cell subtype annotations. We analyzed the attention scores obtained from moSCminer for GSE136718 and GSE154762 datasets and identified the top 30 features with the highest scores from each omics for further examination (Table S6). To assess the relevance of these features to cell subtype classification within cells, we compared the normalized abundance differences between the cell subtypes. We conducted one-way analysis of variance (*Lix, Keselman & Keselman, 1996*) to test the statistical significance of the subtype differences. The results (Fig. 3) revealed that features with the highest attention scores exhibited significant differences among subtypes, with a $p$-value $< 0.01$, providing evidence that moSCminer can identify features highly relevant for distinguishing cell subtypes within cells.

Moreover, the overlap of the top 30 feature lists with the highest attention scores from each omics for the GSE136718 dataset is depicted in the Venn diagram (Fig. 4). This

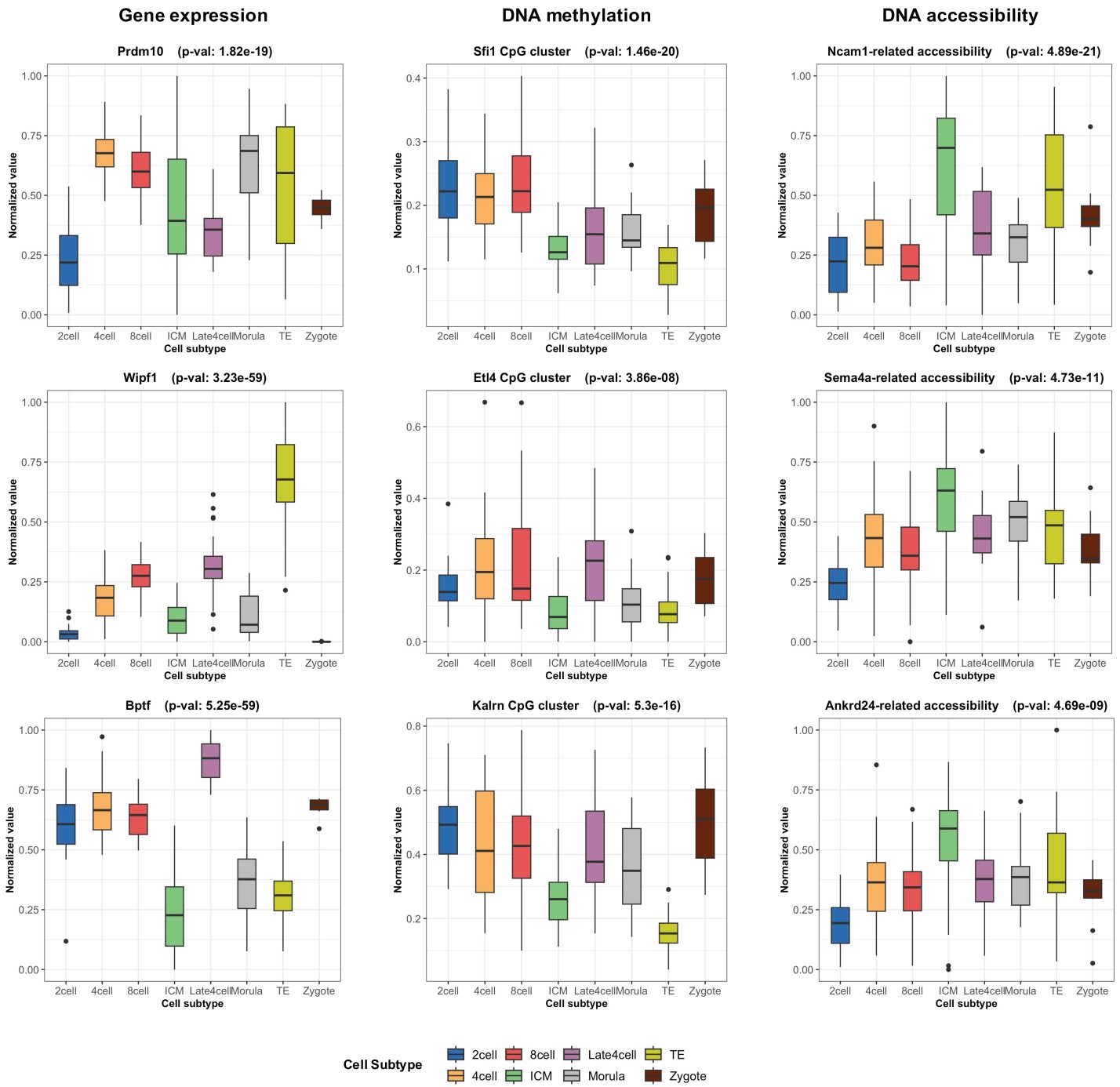

**Figure 3** Normalized abundance difference between the cell subtypes for the top three features from each omics of **GSE136718** dataset showing the top average attention scores.                               

illustrates that moSCminer identifies a subset of features common to multiple omics, reinforcing their potential as robust marker candidates for cell subtype classification.

To further assess the biological relevance of the top 30 features from each omics for each cell type, we conducted a manual literature review. During the preprocessing step, as omics
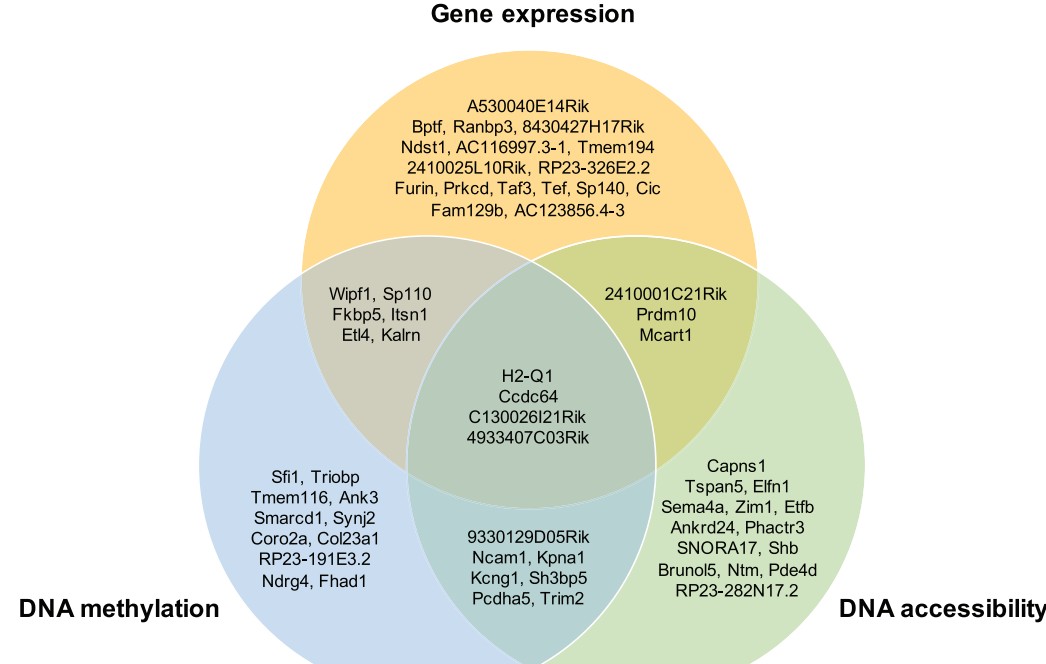

**Gene expression**

A530040E14Rik
Bptf, Ranbp3, 8430427H17Rik
Ndst1, AC116997.3-1, Tmem194
2410025L10Rik, RP23-326E2.2
Furin, Prkcd, Taf3, Tef, Sp140, Cic
Fam129b, AC123856.4-3

Wipf1, Sp110
Fkbp5, Itsn1
Etl4, Kalrn

2410001C21Rik
Prdm10
Mcart1

H2-Q1
Ccdc64
C130026I21Rik
4933407C03Rik

Sfi1, Triobp
Tmem116, Ank3
Smarcd1, Synj2
Coro2a, Col23a1
RP23-191E3.2
Ndrg4, Fhad1

Capns1
Tspan5, Elfn1
Sema4a, Zim1, Etfb
Ankrd24, Phactr3
SNORA17, Shb
Brunol5, Ntm, Pde4d
RP23-282N17.2

9330129D05Rik
Ncam1, Kpna1
Kcng1, Sh3bp5
Pcdha5, Trim2

**DNA methylation**

**DNA accessibility**

**Figure 4 Venn diagram showing the overlap of the top 30 feature lists having the highest attention scores from each omics for GSE136718 dataset.**

features were transformed into a gene-based matrix for multi-omics integration, we examined whether any features exhibited high attention scores in all three omics types. In the case of the GSE136718 dataset, we identified four gene-based features that overlapped among the top 30 features from each omics: H2-Q1, Ccdc64, C130026I21Rik, and 4933407C03Rik (Fig. 4). H2-Q1 gene was reported to be transcribed in the brains of the E14.5 embryos in mice and is also expressed in the adult brain, suggesting a potential functional role in both adult and embryonic brains (*Ohtsuka et al., 2008*). In case of Ccdc64, a synonym of Bicdr1 gene, high expression levels of Ccdc64 at early embryonic stages inhibit neuritogenesis and decrease during embryonic development, thereby controlling neuronal differentiation (*Schlager et al., 2010*).

We also found evidence that other features not in the overlap have the potential to be cell markers: Prdm10 is known to support cell growth and survival during early embryonic development (*Han et al., 2020*). Bptf regulates genes and signaling pathways essential for the development of key tissues in the early mouse embryo (*Landry et al., 2008*). TAF3 and Shb are reported to be involved in embryonic stem cell differentiation (*Kriz et al., 2006*; *Liu et al., 2011*).

For GSE154762 dataset, we did not find overlapping features in any of the three omics datasets. However, several features from each omics dataset demonstrated biological relationships with human oocytes and somatic cells: CAMK1D is reported as a potential follicular cell biomarker correlated with oocyte quality, embryo competence, and pregnancy outcome (*Yerushalmi et al., 2014*). TMOD3 plays a crucial role in oocyte

maturation by controlling the density of the cytoplasmic actin mesh (*Jo et al., 2016*). MYO5B may play an important role in actin cytoskeleton remodeling during oocyte maturation (*Jia et al., 2022*). TMEFF2 was studied for its upregulation in early oocyte development in the primordial and primary follicle stages (*Yu et al., 2020*). These findings illustrate that moSCminer can provide cell marker candidates with the potential to serve as markers for cell subtype annotations.

## DISCUSSION AND CONCLUSIONS

In this study, we introduced an innovative omics-level attention-based framework for cell subtype prediction using single-cell multi-omics datasets. Our approach involved several critical steps, including data preprocessing, feature selection, and the application of an omics-level attention mechanism. These steps were designed to harness the power of multi-omics data and improve cell subtype classification accuracy.

Our preprocessing step involved transforming each omics dataset into a gene-based matrix. This conversion allowed us to identify features with biological relationships based on promoter and transcription information, facilitating effective multi-omics integration. The subsequent application of the omics-level attention mechanism was a key aspect of our approach. By transforming features in each omics dataset into new representations that captured their relative importance weights, we enabled the model to make more informed cell subtype predictions. These transformed representations were then concatenated and used as input for the fully connected layers, with the final cell subtype predictions generated using the softmax function.

The performance of our proposed model was evaluated by comparing it to baseline classifiers using real-world datasets. Through five-fold cross-validation, our model consistently outperformed all other methods, demonstrating robust and superior classification performance. These results underscored the effectiveness of our approach in accurately predicting cell subtypes.

Furthermore, we conducted experiments to investigate the impact of the omics-level attention module and multi-omics integration. Our findings confirmed that the integration of gene expression, DNA methylation, and DNA accessibility data, coupled with the omics-level attention mechanism, significantly improved cell subtype prediction accuracy. This validation further highlighted the advantages of our approach in handling complex multi-omics datasets.

In our experiments, the moSCminer was tested using single-cell multi-omics datasets, which comprised two or three distinct omics profiles. However, it is essential to note that our proposed method is not limited to only three omics types. moSCminer allows its application to multi-omics datasets with varying numbers of omics data. This flexibility is achieved by omics-level attention, by employing a self-attention module for each omics dataset, which generates new representations based on the learned relative importance of features. Subsequently, these features are concatenated and used as input for cell subtype classification. An extensible version of moSCminer can be accessed at https://github.com/joungmin-choi/moSCminer. In addition, moSCminer is originally proposed for cell subtype classification based on single-cell multi-omics dataset integration and provides the

best predictions when trained to learn representations from multiple features grounded in biological connections. However, recognizing the challenge posed by the lack of single-cell multi-omics datasets, we conducted an evaluation of moSCminer's performance using single omics for cell subtype classification, along with baseline methods. Through five-fold cross-validation for each single omics profiles utilizing the GSE140203 dataset, moSCminer consistently outperformed the other baseline methods in cell subtype classification (Table S7). But still, moSCminer achieved the best prediction performance of 0.983 when using multi-omics dataset to classify cell subtypes aligning with our original purpose.

One notable contribution of this study was the development of moSCminer, a user-friendly web-based platform connected to a cloud system. This platform was designed to address common challenges faced by researchers, such as the complexity of software installation and scalability limitations associated with deep learning tools. moSCminer not only simplifies the user experience but also offers easy accessibility to our model. With its intuitive interfaces and visualization capabilities for prediction results, moSCminer provides a practical solution for researchers seeking efficient cell subtype annotation.

In summary, our proposed model represents a significant advancement in single-cell omics studies. By effectively addressing cellular heterogeneity and providing accurate cell subtype annotation, we believe that our approach will contribute to the advancement of research in this field. We believe that the proposed model will help to improve the single-cell omics studies resolving cellular heterogeneity, providing accurate cell subtype annotation.

### Funding

This work was supported by the National Research Foundation of Korea (NRF) grant funded by the Korea government (MSIT) (No. 2021R1F1A1050707), the Bio & Medical Technology Development Program of the National Research Foundation (NRF) funded by the Korean government (MSIT) (2019M3E5D3073365), and by the Agenda Project of the Rural Development Administration, Republic of Korea, under Grant PJ0143072019. The funders had no role in study design, data collection and analysis, decision to publish, or preparation of the manuscript.

### Grant Disclosures

The following grant information was disclosed by the authors:
National Research Foundation of Korea (NRF) grant funded by the Korea government (MSIT): 2021R1F1A1050707.
Bio & Medical Technology Development Program of the National Research Foundation (NRF) funded by the Korean government (MSIT): 2019M3E5D3073365.
Agenda Project of the Rural Development Administration, Republic of Korea: PJ0143072019.

## Competing Interests

The authors declare that they have no competing interests.

## Author Contributions

- Joung Min Choi conceived and designed the experiments, performed the experiments, analyzed the data, prepared figures and/or tables, authored or reviewed drafts of the article, developed web-based platform and connected the cloud system, and approved the final draft.
- Chaelin Park conceived and designed the experiments, performed the experiments, prepared figures and/or tables, developed web-based platform and connected the cloud system, and approved the final draft.
- Heejoon Chae conceived and designed the experiments, authored or reviewed drafts of the article, supervised the whole manuscript, and approved the final draft.

## Data Availability

The data is available at NCBI GEO: GSE154762, GSE136718, and GSE140203.

The analysis source code is available at GitHub and Zenodo:

- https://github.com/joungmin-choi/moSCminer.

- Joung Min Choi. (2024). joungmin-choi/moSCminer: moSCminer v1.0.0 (v1.0.0). Zenodo. https://doi.org/10.5281/zenodo.10633241.

moSCminer is available at: http://203.252.206.118:5568/.

## Supplemental Information

Supplemental information for this article can be found online at http://dx.doi.org/10.7717/peerj.17006#supplemental-information.

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
