# Peer review of "moSCminer: a cell subtype classification framework based on the attention neural network integrating the single-cell multi-omics dataset on the cloud"

_PeerJ, doi:10.7717/peerj.17006_

## Round 0.1 · original submission · Major Revisions

Please revise the manuscript by following the reviewers' comments.

Reviewer 1 ·

Basic reporting

Considering the shortcomings of traditional methods and single-omics in cell subtype classification, the authors develop moSCminer, a novel framework for classifying cell subtypes that harnesses the power of single-cell multi-omics sequencing datasets through an attention-based neural network operating at the omics level. This manuscript could be accepted if the following points are taken care of.

Experimental design

no comment

Validity of the findings

1. In the part “performance evaluation with the baseline methods”, the author has selected too few evaluation metrics. In addition to accuracy and F1-score, the author should provide some other evaluation metrics to demonstrate the performance of the classifier.
2. In the part “DISCUSSION AND CONCLUSIONS”, the author should discuss whether the number of multiple omics is extensible.
3. In the part “Cell subtype prediction improvement by multi-omics integration”, in a single omics experiment, the accuracy and F1 scores of experiments based on DNA accessibility are low relative to other omics, and authors may consider other DNA accessibility datasets to illustrate this situation.

Additional comments

Considering the shortcomings of traditional methods and single-omics in cell subtype classification, the authors develop moSCminer, a novel framework for classifying cell subtypes that harnesses the power of single-cell multi-omics sequencing datasets through an attention-based neural network operating at the omics level. This manuscript could be accepted if the following points are taken care of.
1. In the part “performance evaluation with the baseline methods”, the author has selected too few evaluation metrics. In addition to accuracy and F1-score, the author should provide some other evaluation metrics to demonstrate the performance of the classifier.
2. In the part “DISCUSSION AND CONCLUSIONS”, the author should discuss whether the number of multiple omics is extensible.
3. In the part “Cell subtype prediction improvement by multi-omics integration”, in a single omics experiment, the accuracy and F1 scores of experiments based on DNA accessibility are low relative to other omics, and authors may consider other DNA accessibility datasets to illustrate this situation.

Reviewer 2 ·

Basic reporting

Choi et al. proposed an attention-based method moSCminer that uses multi-omics data for classifying cell subtypes. Choi et al. selected a few widely used conventional classification methods for benchmarking and showed superior performance of moSCminer. The authors also implemented their tools as cloud-based web tool, which could be more accessible.

Experimental design

The authors did well documentation and explaining on model. Method section is also clear to read.

Validity of the findings

1. The only two datasets used in this study have a very small number of cells. Nowadays, a single single-cell omics dataset can easily surpass one hundred thousand cells. The demonstration with “small” single-cell datasets is not convincing to show the proposed method can handle different scenarios of single-cell data analysis.
2. The authors claimed that moSCminer achieved better accuracy than other baseline methods. However, all RF, SVM and LR have over 0.95 accuracy. Consider the testing data set used for evaluation is small, it is hard to justify that moSCminer indeed has great performance.
3. Choi et al. claimed that moSCminer aims for cell subtype classification. However, both datasets used in this study have less than 10 cell types, and they are all major cell types. Different major cell types are distinctly different. Thus, classification of major cell types is considered relatively easy. However, different cell subtypes, for example different T cell subtypes, may only present some slight differences. Subtype classification can be very challenging. The two datasets used here may not be qualified to support that moSCminer is capable of cell subtype classification.
4. It is nice to have a tool that makes full use of multi-omics data for classification. However, most existing single-cell datasets don’t have different omics available for use. There is no benchmark in this study to show if moSCminer (with attention mechanism) would still be superior to baseline methods (RF, SVM, and etc.) if only using single omics.
5. In Table 2, single-cell gene expression data alone can be informative for accurate cell-type predictions. As each omics was processed by neural network independently and the classifier used the concatenated latent embeddings of these 3 omics, it is counterintuitive that “gene+methyl+acc” got boosted performance while both “gene+methyl” and “gene+acc” dragged the accuracy down. A possible explanation could be the test set is too small to be quantified for evaluating model. 0.2 of 210 cells (8:2 ratio and 5-fold cross-validation) has only 42 cells for evaluation.
6. In Figure 2, it is odd to see Accuracy and F1 have very similar mean or median (There is no legend to describe what the horizontal lines stand for in each boxplot). It could be possible that calculations of both metrics were averaged globally. Model can perform well for the major cell type and don’t necessarily need to predict minor cell type accurately. Again, the testing set is very small, especially in the case of GSE136718. It is skeptical that the benchmark with baseline methods demonstrates moSCminer is superior to baseline methods. There is lack of supplementary report to show the prediction in each cell type.

Reviewer 3 ·

Basic reporting

(1) The reason that attention neural network was used for the framework should be explained. (2) It is suggested to cite some recently published references related to this article to increase its relevance to the research topic. For example, https://doi.org/10.1038/s41421-022-00415-0

Experimental design

The colors used in Figure 3 are suggested to be accompanied by annotations.

Validity of the findings

It is suggested to expand the content of "The effectiveness of omics-level attention" and "Cell subtype prediction improvement by multi-omics integration" in the Results section to enrich the manuscript.

Additional comments

The manuscript provides an innovative omics-level attention-based framework for cell subtype prediction using single-cell multi-omics datasets. The manuscript serves as a valuable resource for researchers in the field of single cell analysis. The notable contribution of this study was the development of moSCminer, a user-friendly web-based platform connected to a cloud system.

---

## Round 0.2 · accepted · Accept

The authors have addressed the comments and concerns of all three reviewers satisfactorily. So, I would like to accept the manuscript for publication in PeerJ.

Reviewer 2 ·

Basic reporting

The revised version provided extra background information about attention-based models and described current research progress in single-cell classification with attention model. I found no problem of reading the article and figures are more clear to understand than previous version.

Experimental design

In the revised version, the author added additional dataset for benchmark. They also included extra evaluation metrics to demonstrate the proposed method outperforms the baseline methods. They answered my concerns in previous reviewing. I don't have new concerns for this revised manuscript.

Validity of the findings

I am convinced by new results the author provided. Especially, individual accuracy for each subtype shows attention-based method could outperform conventional machine learning methods.